# Paralytic Impact of Centrifugation on Human Neutrophils

**DOI:** 10.3390/biomedicines10112896

**Published:** 2022-11-11

**Authors:** Tobias Hundhammer, Michael Gruber, Sigrid Wittmann

**Affiliations:** Department of Anesthesiology, University Hospital Regensburg, 93042 Regensburg, Germany

**Keywords:** neutrophils, PMNs, isolation, centrifugation, *g*-time, *g*-forces, impairment, neutrophil functions

## Abstract

Centrifugation is a common step in most of the popular protocols for the isolation of neutrophils from whole blood. Inconsistent results from previous studies on neutrophils may originate from an underestimation of the centrifugation effect, as in consequence impaired, not native cells, being investigated. We hypothesize, that centrifugation significantly impairs major neutrophil functions. However, there is no data yet whether the application of *g*-force itself or the product of *g*-force and duration of centrifugation (=“*g*-time”) defines the impact on neutrophils. Neutrophils were isolated from whole blood via centrifugation with different *g*-times and subsequently analyzed via live cell imaging for migration, as well as via flow cytometry for oxidative burst and surface antigen expression. Chemotactic migration was significantly reduced with increasing *g*-time. Oxidative burst decreased likewise the higher the *g*-time applied. Expression of CD11b was no longer upregulated in response to an n-formylmethionine-leucyl-phenylalanine (fMLP) stimulus in neutrophils having experienced high *g*-time during the isolation process. We conclude that centrifugation “paralyzes” neutrophils in the form of a significant decrease in functionality. Future investigations on neutrophil granulocytes should reduce the *g*-time load as far as possible.

## 1. Introduction

Neutrophils, representing a group of cell types called polymorphonuclear leukocytes (PMNs), alongside eosinophils and basophils, are the most common circulating leukocytes in human blood [1]. Since neutrophils are by far the most abundant granulocytes [1], “PMNs” will be used synonymous to “neutrophils” in this article. Their variety of antimicrobial weapons makes them an indispensable part of the innate immune system [2] as well as a highly interesting field of medical research.

Due to the major interest in neutrophils as subjects of studies on immunity in general, appropriate supply must meet the needs of laboratory investigations. Neutrophils are impossible to cultivate, and neutrophil-like cell lines do not reproduce all neutrophil characteristics to a satisfying extent [3]. Thus, isolation of PMNs from fresh whole blood and their immediate use has been the modus operandi for most of the studies on neutrophils. A multitude of isolation methods has been established over the years, roughly discriminable into two major categories: the relatively simple but rather time-consuming physical methods such as density gradient separation, versus the more advanced, but quite expensive separation methods, such as magnetic-activated cell sorting (MACS) and fluorescence-activated cell sorting (FACS).

Density gradient separation primarily relies on the differences in intrinsic density between different cell populations: First, a fluid containing multiple densities has to be filled into a container, with the fluid of least density located on top. After layering a sample of whole blood upon the separation fluid, different cell types will start sinking—forced by gravity (1*g*)—until they reach a layer identical to their own density, their so called “isopycnic point” [4]. From there on, those cells do not sink any further and can be extracted. While the simplest forms of such media mainly allow separation of the denser red blood cells (density ρ = approx. 1.110 g/mL) from the less dense white blood cells (ρ = approx. 1.080 g/mL) [4], more advanced types (e.g., Lympho Spin^®^ Medium and Leuko Spin^®^ Medium, both pluriSelect life science, Leipzig, Germany) even permit sorting of different leukocyte populations [5]. Although gravitation theoretically imposes enough force (1*g*) on the blood cells for them to descend, this process may last up to a few hours [4], causing a significant delay in the study’s setting. This problem can be solved by centrifuging the tube containing the samples [4], as proposed in various protocols for neutrophil isolation from whole blood [6,7]. However, the high relative centrifugal forces (RCF, in this context synonymous with “*g*-force”) required for this separation could potentially damage the targeted cells.

Magnetic-activated cell sorting (MACS) is a form of positive selection of certain cell populations out of cell suspensions or whole blood. In a first step, magnetic beads bearing e.g., anti-CD15 antibodies are incubated in a tube with whole blood, where they bind to the CD15-positive neutrophils. Next, inducing a magnetic field around the sample leads the bead-bound neutrophils to attach to the tubes’ wall. After eluding cells not bound to the containers’ wall and then removing the magnetic field, the remaining target cells can be harvested [8].

Fluorescence-activated cell sorting (FACS) allows the automatic withdrawal of fluorescently labeled target cells from a cell suspension. However, cell sorters in this category may not be available to many laboratories, as they are associated with major acquisition and maintenance costs.

To date, a variety of studies regarding the impact of isolation methods on neutrophils have been published. Some of those have focused on the different density gradient media used for isolation [9,10]. Other publications aimed to compare density gradient separation methods to advanced methods such as MACS or FACS regarding the potential manipulation of PMNs [11,12,13].

However, none of the pre-existing studies has—to our knowledge—focused on the impact of centrifugation itself on neutrophil functions. In fact, while centrifugation is the main step in density gradient separation protocols, it is also commonly featured in MACS and FACS separation methods, for example, during sample preparation or washing steps [13,14,15,16].

As intriguing as neutrophils are as a subject of research, their short half-life [17] and low threshold for activation [9,18] account for major influencing factors. Investigation on PMNs usually requires isolated cells, still as native as possible. For studies aiming to detect changes in migration, expression of surface markers or granule contents caused by biological or therapeutic treatments, isolation stress might be detrimental.

This study aims to address these complications and demonstrate the impairment of neutrophil functions among density gradient separation methods compared to a simple 1*g* sedimentation.

To minimize external influences as far as possible, separation methods relying on positive selection through ligand-receptor interactions with PMNs (e.g., MACS) had to be excluded from this study.

We hypothesize that *g*-forces combined with the duration of centrifugation significantly impair major neutrophil functions. To test our hypothesis, we aim to answer the following questions in the course of this study:

Are there significant differences detectable regarding motile activity, surface antigen expression and oxidative burst of neutrophils when comparing even low, but different, relative centrifugal forces (1*g*, 10*g*, 20*g*, 30*g*, 47*g*, 756*g*) used for density gradient centrifugation at identical duration? Does the duration of centrifugation influence the cellular parameters at one distinct *g*-force?

## 2. Materials and Methods

### 2.1. Study Plan

To assess the question of whether neutrophil functions are impaired after isolation from whole blood via centrifugation, two different setups are used in this study (Figure 1).

### 2.2. Introduction of “g-Time” as a New Parameter of Centrifugation

For our investigation on the impact of centrifugation on neutrophils, we decided to implement a new parameter called “*g*-time”. This variable factors in the two main constants of centrifugation: The *g*-force (RCF) applied and centrifugation duration (CT). Mathematically, *g*-time is defined as:*g*-time [*g*s] = (RCF [*g*] × CT [s])(1)
and has the unit “*g*-seconds [*g*s]” or “kilo-*g*-seconds [k*g*s]” (1 k*g*s = 10^3^ *g*s) for high values, respectively. *g*-time can be applied as a parameter under the condition that the RCF used for centrifugation is >1*g*. When sedimentation is performed at 1*g*, this is referred to as “control” or “1*g*” in the context of *g*-time.

### 2.3. Blood Withdrawal and Sample Preparation

Each trial required blood donation from a volunteer, who gave their informed consent according to the positive vote of the local ethics committee (16-101-0322). Blood withdrawal was performed using one to four lithium heparin-anticoagulated blood collection tubes as well as one serum clot activator tube (both SARSTEDT AG & Co. KG, Nuembrecht, Germany).

The serum tube was centrifuged at 708 k*g*s (1180*g* for 10 min) at room temperature. Afterwards, the supernatant containing solely blood serum could be extracted and stored in a 2 mL centrifugation tube for further usage.

### 2.4. Isolation of Neutrophils from Whole Blood

Lithium heparin-anticoagulated blood was separated into multiple 2 mL centrifugation tubes for further preparation. PMNs were isolated from whole blood by applying different RCFs to the whole blood samples and collecting the leukocyte-rich supernatant afterwards (Figure 2).

The control was set as 1*g*, meaning whole blood was allowed to sediment until gravity (1*g*) induced enough separation of red blood cells so that the leukocyte-rich supernatant could be withdrawn. For the other samples, sedimentation was accelerated by centrifugation.

In order to provide equal amounts of contact time between the neutrophils and other blood components for each sample, centrifugation was started not until the 1*g*-method had produced a satisfying volume of supernatant (lasting an average of 45 min). It was necessary to resuspend the samples intended for centrifugation before loading them into the different centrifuges, since a certain amount of sedimentation had already taken place after 45 min.

For analysis of neutrophil migration via live cell imaging, four different *g*-forces were compared to the 1*g* control: 10*g*, 20*g*, 30*g* and 47*g*. For isolation with 1*g*, 10*g*, 20*g* and 47*g*, 2 mL samples of whole blood were pipetted into 2 mL reaction tubes (Greiner Bio-One International GmbH, Kremsmuenster, Austria). Isolation with 30*g* required the 2 mL sample of whole blood being filled into a 15 mL CELLSTAR^®^ tube (Greiner Bio-One International GmbH). The 47*g* sample was centrifuged with Heraeus™ Megafuge™ type 1.0 R (Thermo Fisher Scientific, Waltham, MA, USA). The 30*g* sample was centrifuged with BioFuge™ Stratos™ centrifuge (Thermo Fisher Scientific). Microstar 17R Microcentrifuge (VWR International, Radnor, PA, USA) was used for centrifugation of the 10*g* and 20*g* samples.

Centrifugation duration (CT) for samples analyzed via live cell imaging varied between 5 and 25 min, depending on the requirements for the respective test series. Accordingly, the *g*-time applied for analysis of chemotactic migration ranged from 6 to 45 k*g*s. For samples analyzed via flow cytometry, centrifugation duration was set to 15 min.

After centrifugation and sedimentation were finished, supernatants were transferred into separate containers.

Examination of antigen expression and oxidative burst included one additional sample, where PMN isolation was performed by density gradient separation with density gradient medium. Here, 2 mL of LeukoSpin^®^ medium (pluriSelect Life Science) were laid into a 15 mL centrifugation tube and then overlaid with 2 mL of PBMC Spin^®^ Medium (pluriSelect Life Science). A 2 mL probe of whole blood was set on top as third layer. Centrifugation was performed at 756*g* for 15 min without braking, using the BioFuge™ Stratos™ (Thermo Fisher Scientific). Afterwards, the granulocyte-rich layer was removed according to the manufacturer’s instructions and transferred into a new container for further usage. This isolation method is referred to as “756*g*” corresponding to “680 k*g*s” in the following, but it is always performed as a density gradient centrifugation with density gradient medium, as described above.

### 2.5. Cell Migration Assay

To induce and observe cell migration, 3D-µ-Slide migration-chambers (Ibidi GmbH, Graefelfing, Germany) were used. One slide contains three separated systems or chambers, each chamber consists of a central channel and a left and a right reservoir (Figure 3).

The three chambers were filled as described in Table 1 and Figure 3b. In this setup, the PMNs—placed in the right reservoir—migrated along the gradient of the chemoattractant n-formylmethionyl-leucyl-phenylalanine (fMLP; Sigma Aldrich, St. Louis, USA), which was brought into the left reservoir. During migration, cells had to pass through the collagen matrix inside the central channel and could thereby be examined via live cell imaging (Figure 3c). Based on preliminary tests (data not shown), where we had already proven that only granulocytes were able to pass the collagen matrix, we were assured that solely this cell type (CD11b^+^; CD62L^+^; CD66^+^) is observed.

Cell observation was performed with a Leica DMi8 microscope and recorded with Leica DFC9000 GT camera (both Leica Microsystems GmbH, Wetzlar, Germany) for 22 h. Both camera and microscope were controlled by Leica Application Suite X software platform, version 3.4.2.18368 (Leica Microsystems GmbH). A stage top incubator consisting of Ibidi Blue Line gas mixer and Ibidi heating chamber (both Ibidi GmbH) was used to keep conditions constant at 37 °C and 5% CO_2_.

### 2.6. Examination of Chemotactic Migration

#### 2.6.1. Schedule of Observation

Chemotactic migration of neutrophils was analyzed using Imaris^®^ Microscopy Imaging Analysis Software (Version 9.0.2, Bitplane, Zurich, Switzerland).

Neutrophils were tracked over the course of 22 h (Figure 4), with one picture (=frame) of each channel being taken every 30 s and one analysis interval consisting of 60 frames (=30 min).

In the first 180 min (3 h), every frame taken was evaluated. From there on, the distance between the analysis intervals was extended, so that from hour 5 of observation on, 60 frames were evaluated only every 5 h. Each analysis interval was labeled after its first frame (=start frame), achieving 10 analysis intervals in total.

#### 2.6.2. Migration Analysis

For migration analysis, we evaluated each eligible frame regarding the parameters shown in Table 2.

For each of the three channels, 50 cells with the highest track length values within each start frame were considered for analysis, which totaled 500 cells per channel for the entire observation period.

Afterwards, the data was exported to IBM^®^ SPSS^®^ statistics software program (version 28.0.0.0, IBM, Armonk, NY, USA). Thereupon, only those cells which exceeded track lengths ≥ 25 µm and track durations ≥ 900 s were taken into consideration for further evaluation. To reach more clarity for our graphics, start frames were categorized into the new variable “observation period”: Start frames 1 to 301 were summed up to “First 3 h”, whereas start frames 601 to 2401 are represented by the term “Last 19 h”.

Combined data from every live cell imaging trial was also exported to an Excel (Microsoft Corporation, Redmond, DC, USA) file and then exported to and analyzed via Phoenix 64 NLME™ software (Build 8.1.0, Certara Inc., Princeton, NJ, USA).

### 2.7. Flow Cytometry

#### 2.7.1. Surface Antigen Expression

Surface antigen expression was examined with PMNs isolated at 9–680 k*g*s (10*g*, 20*g*, 47*g*, 756*g* each for 15 min) as well as with the 1*g* control group. Briefly, 200 µL of the isolated leukocyte-rich supernatant were laid into a 2 mL centrifugation tube; 590 µL of RPMI 1640 medium (PAN-Biotech GmbH, Aidenbach, Germany), 200 µL of autologous serum and 10 µL of 1 µM fMLP (Sigma Aldrich) were added. From this leukocyte-suspension, samples were withdrawn before incubation was started (t = 0) as well as after incubation for 22 h (t = 22) at 37 °C in a HeraCell™ 150i CO_2_-Incubator (Thermo Fisher Scientific).

Expression of surface antigens was examined by immunostaining with PE-conjugated anti-CD11b (ICRF44, BioLegend, San Diego, CA, USA), FITC-conjugated anti-CD62L (DREG-65, BioLegend) and APC-conjugated anti-CD66b (G10F5, BioLegend). The immunostaining process involved two centrifugation steps with a combined *g*-time load of 209 k*g*s, which were both performed at 4 °C immediately before flow cytometric measurement.

#### 2.7.2. Oxidative Burst

Oxidative burst was investigated by flow cytometry directly after the neutrophil isolation process, which included *g*-times of 9–680 k*g*s (10*g*, 20*g*, 30*g*, 47*g* and 756*g* each for 15 min) and the 1*g* control group. Samples were withdrawn from the not further treated leukocyte-rich supernatants gained in the isolation process. Burst was stimulated by fMLP (10 µM, Sigma Aldrich) in combination with tumor necrosis factor alpha (TNF-α; 1 µg/mL, Thermo Fisher Scientific), or otherwise with phorbol-12-myristate-13-acetate (PMA; 10 µM, Sigma Aldrich), which was set as positive control. ROS-production was detected by oxidation of non-fluorescent Dihydrorhodamine (DHR) 123 (10 µM, Thermo Fisher Scientific) to fluorescent Rhodamine 123. This process has already been described in a more detailed way in previous publications of our department [19,20].

#### 2.7.3. Flow Cytometric Measurement and Software Data Analysis

Flow cytometry was performed by using a FACSCalibur™ flow cytometer (BD corporate, Franklin Lakes, NJ, USA) and CellQuest Pro software™ (version 5.2, BD corporate). Surface antigen expression and oxidative burst were investigated with cell counts of 10.000 cells per sample. Neutrophils were identified by their typical patterns displayed in forward-scatter (FSC) and side-scatter (SSC) light. Data analysis was executed with FlowJo™ (V10.0.7, BD corporate) analysis software.

For examination of surface antigen expression, data had to be further treated after being exported to an Excel file. First, median fluorescence intensity (MFI) levels at t = 0 were set as default value for each individual *g*-time group. MFI values at t = 24 were then converted to “relative change in expression” in relation to the levels at t = 0. Afterwards, “relative change in expression” values for each surface antigen, which were only explainable with errors in sample preparation or in the measurement process, had to be excluded. The threshold value for this elimination was set as a greater-than-10-fold deviation from the mean value of the other measurements for the same *g*-time in the test series. In case this threshold was exceeded for one of the analyzed surface antigens, the whole measurement was excluded from the analysis, so that values for both other surface antigens were likewise ruled out. After this first preparation step had been performed, the data was exported to IBM^®^ SPSS^®^ statistics software program (version 28.0.0.0, IBM). Before examining statistical differences between the different groups, statistical outliers were excluded.

### 2.8. Statistical Evaluation

Data from neutrophil migration, surface antigen expression and oxidative burst were exported to Excel (Microsoft Corporation) files. IBM^®^ SPSS^®^ statistics software program (version 28.0.0.0, IBM) was used for further statistical evaluation. Every result was tested for Gaussian distribution with the Kolmogorov–Smirnov test.

In case of normal distribution, data was analyzed via one-way ANOVA. Before performing post hoc tests, Levene’s test was used to indicate homogeneity of variance. In case of homogeneity, Bonferroni’s test was used for post hoc analysis. When no homogeneity of variance existed, Games–Howell’s test was performed for post hoc analysis instead.

In the case of the data not showing normal distribution, the Kruskal–Wallis test was performed as non-parametric test and Bonferroni-correction was applied.

Differences between analyzed groups in graphics are marked as: ns = not significant; * *p* < 0.05; ** *p* < 0.01; *** *p* < 0.001. Small circles in graphics mark outliers between 1.5 × the interquartile range (IQR) and 3 × IQR. Small triangles in graphics mark outliers of >3 × IQR.

## 3. Results

Whole blood was withdrawn from 18 donors in total (10 males, 8 females) with a median age of 23 years.

### 3.1. Chemotactic Migration

For migration analysis inside the µ-Slides, in each trial we compared neutrophils isolated by centrifugation with two different *g*-times to the 1*g* control.

The mean track length of PMNs isolated at 1*g* was set as a default value for each analysis interval, and mean track lengths of cells isolated at the different *g*-times under investigation were expressed as “Relative Deviation from Control’s Track Length (rTL)” for each analysis interval.

#### 3.1.1. Impact of g-Force on Neutrophil Migration

In the first step, we examined neutrophil migration with cells isolated at different *g*-forces but the same centrifugation duration, which was set to 10 min (Figure 5).

No significant difference could be found between cells isolated at 6 k*g*s and the control cells. Relative track lengths of cells isolated at 12, 18 and 28 k*g*s were significantly reduced compared to those of neutrophils having experienced 6 k*g*s (*p* < 0.05 for 18 k*g*s, *p* < 0.001 for 12 and 28 k*g*s) as well as to those of the control cells (*p* < 0.001 for each).

#### 3.1.2. Effect of Centrifugation Duration on Neutrophil Migration

To exclusively test the effect of centrifugation duration, whole blood was centrifuged at the same RCF, but for different time spans. Afterwards, neutrophil migration was examined with those cells and compared to 1*g*-control cells as usual.

Examination of different centrifugation durations for 20*g* (Figure 6a) showed a small but significant (*p* < 0.05) difference in migration between the 1*g* control cells and neutrophils centrifuged at 6 k*g*s (20*g* for 5 min). Significant impairment (*p* < 0.001) of neutrophil migration was also displayed in PMNs centrifuged at 18 k*g*s (20*g* for 15 min) compared to control cells. While track lengths of 6 k*g*s cells were only reduced by a mean value of 8.7% versus the 1*g* control, reduction of track lengths with 18 k*g*s cells was 53%. This difference was significant as well (*p* < 0.05).

When investigating different CTs at 30*g* (Figure 6b), the track length was negatively correlated to the time of exposition. Not only were there significant differences in track length between the 1*g* control and all 30*g* samples (*p* < 0.05 for CT = 5 min; *p* < 0.001 for CT = 25 min), PMNs having experienced 45 k*g*s (=30*g* × 25 min) again displayed a significantly (*p* < 0.001) impaired chemotactic migration compared to those centrifuged at 9 k*g*s (=30*g* × 5 min).

#### 3.1.3. Combination of *g*-Force and Centrifugation Duration to “*g*-Time”

The above-presented results indicate that both the *g*-force applied as well as the duration of centrifugation significantly influence the neutrophils’ chemotactic ability. For the following examinations, splitting of both factors is avoided and instead the combined parameter “*g*-time” is used to compare neutrophil migration.

When comparing different *g*-times applied to neutrophils compared to the 1*g* control cells, a continuous decrease in track lengths can be seen the higher the *g*-times applied (Figure 7a). While differences with cells isolated at 6 k*g*s and at 9 k*g*s did not reach statistical significance, chemotactic migration was significantly reduced once neutrophils had experienced more than 10 k*g*s (*p* < 0.001 for each group). The steepest decline in track length could be seen between 9 and 28 k*g*s.

Comparison of *g*-time intervals with each other affirmed this impression (Figure 7b). *g*-time < 10 k*g*s did not significantly impact neutrophils’ ability of chemotactic migration, but cells isolated at 10–19 k*g*s were only able to reach 58% of the track lengths the “<10 k*g*s”-cells did (*p* < 0.001). PMNs having experienced 20–29 k*g*s were even further impaired and were only able to reach 51% of the “10–29 k*g*s”-cells’ track lengths (*p* < 0.01). Lowest track length values were recorded with neutrophils centrifuged at >30 k*g*s.

Performing an inhibitory dose–response model for the effect (response) of *g*-time (dose) on track length of neutrophils produced a sigmoidal curve (Figure 8a). A zero effect niveau (E_0_) was calculated close to 0% relative deviation from control’s track length. The decline in relative track length occurred between 10 k*g*s and 30 k*g*s. E*g*t_50_, which marks the *g*-time, at which half of the maximum rTL-decrease is reached, was calculated to be at 20.0 k*g*s, followed by the maximum effect level (E_max_) of −80%. The E*g*t_50_-value is already reached at relatively small *g*-forces and short centrifugation duration, as Figure 8b shows: For example, centrifugation at 50*g* may last approximately 7 min to produce a *g*-time load of 20.0 k*g*s. However, when centrifuging at 320*g*, 20.0 k*g*s are already reached after just 1 min.

### 3.2. Oxidative Burst

To determine whether acceleration forces may also impair the PMNs’ microbicide mechanisms, we performed an experimental series measuring oxidative burst of neutrophils after isolation with different *g*-times (9, 18, 27, 42, 680 k*g*s). Similar to the analysis of chemotactic migration, median fluorescence intensity (MFI) values of the 1*g* sample were set as a control value (“0”). For the other samples, MFI values are expressed as “Relative Deviation from 1*g* Control [%]”.

When stimulated with TNF-α and fMLP, oxidative burst decreased continuously with increasing *g*-time (Figure 9a). The steepest decline occurred between 9 and 27 k*g*s. The lowest values measured corresponded to isolation with 27 and 42 k*g*s, with both groups differing significantly from the 1*g* control (*p* < 0.01 for both). Isolation via density gradient separation at 680 k*g*s was likewise associated with less oxidative burst, although those values did not reach significant difference to the 1*g* control and evened out at a level similar to centrifugation at 18 k*g*s.

In the case of oxidative burst being stimulated with PMA (Figure 9b), which was used as positive control, burst levels of 27 and 42 k*g*s cells were lower than with the 1*g* neutrophils. Although the differences in this case did not reach statistical significance, those results displayed a similar trend as the ones after stimulation with TNF-α and fMLP. However, in contrast to stimulation with TNF-α and fMLP, neutrophils isolated via density gradient centrifugation at 680 k*g*s displayed oxidative burst levels, which were even significantly higher (*p* < 0.05) than those after isolation with 1*g*.

### 3.3. Expression of Surface Antigens CD11b, CD66b, CD62L

The impact of centrifugation on PMNs’ expression of surface antigens CD11b, CD62L and CD66b was examined via flow cytometry directly after the isolation process as well as after 22 h of incubation at 37 °C with fMLP.

At first, we analyzed the median fluorescence intensities (MFIs) of the three stains conjugated to the antigens’ respective antibodies in order to assess antigen expression at both measurement periods (t = 0; t = 22). When examining neutrophils immediately after isolation from whole blood (t = 0), expression of CD11b and CD66b antigens did not significantly differ between PMNs isolated via centrifugation and the control group (1*g*). Moreover, no significant differences could be found between the different *g*-times applied in the centrifugation process (9, 18, 42, 680 k*g*s) at this point of time. However, analysis of surface antigens after 22 h of incubation with fMLP showed major differences in contrast to the previous measurement (Figure 10). CD11b expression decreased significantly (*p* < 0.001) with the higher the *g*-time applied during the isolation process (Figure 10a). Isolation with 42 k*g*s (*p* < 0.01), as well as isolation via density gradient centrifugation at 680 k*g*s (*p* < 0.001) in particular, led to significantly lower levels in CD11b expression after 22 h compared to the 1*g* control. Considering the relative change in CD11b expression before (t = 0) and after (t = 22) incubation with fMLP, only the PMNs of the control group, as well as those isolated at 9 k*g*s, were able to answer said chemoattractant stimulus with an upregulation of CD11b surface antigen expression (Figure 10b). Neutrophils isolated at 42 k*g*s (*p* < 0.05) or 680 k*g*s (*p* < 0.05), however, did not show this ability as CD11b expression levels after 22 h went below those measured before incubation (t = 0).

Examination of CD66b expression showed a similar overall picture. Control cells as well as neutrophils centrifuged at 9 and 18 k*g*s each displayed a major increase in CD66b surface antigen expression after 22 h of incubation with fMLP. Cells having experienced a *g*-time of 42 k*g*s provided upregulation of CD66b expression to a lesser—but not significantly different—degree. However, PMNs isolated at 680 k*g*s did not show increased CD66b levels after incubation, which marked a significant difference (*p* < 0.05) to the 1*g* control (Figure 10c,d).

In contrast to the abovementioned markers, expression of CD62L surface antigen decreased consistently across all separation methods after 22 h of incubation and therefore did not show any correlation to the *g*-time applied in the isolation process (data not shown).

## 4. Discussion

The impact of different separation methods on neutrophil migration has been the subject of multiple studies in the past. Some researchers have compared different media used for density gradient separation [9,10,11]. Other studies again focused on the differences between density gradient separation and other methods like immunomagnetic neutrophil isolation (MACS) [8,13,21]. For instance, Blanter et al. reported, that chemotactic migration was comparable between density gradient versus immunomagnetically isolated PMNs [13]. Centrifugation steps have been involved in all of those studies, be they as a main part of the isolation process or just during cell-washing steps. However, to our knowledge, none of the existing studies comparing different isolation methods has considered (even short or slow) centrifugation as possible factor for neutrophil impairment.

Our data clearly indicate that neutrophil migration is impaired in PMNs isolated from whole blood via any centrifugation. When whole blood samples were centrifuged with different RCFs but equal centrifugation duration, track lengths from high-RCF cells were significantly decreased compared to low-RCF and control cells. When centrifugation was performed with equal RCF but different centrifugation durations, high-CT cells had lower track length values than low-RCF cells. In our opinion, the impairment of neutrophil migration by centrifugation is a product of the two factors *g*-force and centrifugation duration, a product which we referred to as “*g*-time”.

Our findings contradict previous studies [13,21], in which density-gradient-separated neutrophils did not show significantly less chemotactic migration than PMNs isolated with other methods.

Following our hypothesis, it would make sense that no major differences in PMN migration between those methods have been found yet, since the key trigger for decreased migration has occurred in each of the compared groups.

The inhibitory dose–response model we performed supports our findings. Considering the E*g*t_50_-value being 20.0 k*g*s and the steepest decline occurring between 10 and 30 k*g*s, we assume that the major effect centrifugation has on neutrophil migration already takes place at small *g*-time-doses. This would be particularly interesting, as most of the existing isolation protocols exceed this threshold by far [8,9,13,22,23,24,25]. However, since rTL values do not further decrease beyond approximately 80 k*g*s, researchers may not have noticed this impairment. The neutrophils they examined still showed a certain amount of migration, although significantly less than more native cells would have.

Similar to chemotactic migration, several researchers have tried to point out the most suitable method to isolate PMNs for analysis of oxidative burst [8,13,21,23]. However, again, no one has yet focused on the impact of centrifugation itself on oxidative burst in neutrophils.

Our study showed a negative correlation of the oxidative burst and the *g*-time, regardless of the stimulant (fMLP or PMA) being used. The only exemption to this was when burst was stimulated with PMA in cells isolated via density gradient separation at 680 k*g*s. In this case, burst levels were even elevated compared to the 1*g* isolated cells.

Located at the neutrophil’s surface, a membrane-bound and G-protein-coupled receptor is responsible for detecting and binding fMLP [26] PMA receptors, however, are located in the cytoplasm of PMNs and are associated with PKC activity themselves [27].

Our results indicate that centrifugation impairs both surface receptor- and intracellular-receptor-mediated ROS production. However, we cannot explain why ROS production is decreased in cells isolated with density gradient media at 860 k*g*s after stimulation with fMLP + TNF-α but is even elevated when burst was induced with PMA.

Again, the lack of pre-existing data on differences in ROS production depending on the neutrophil isolation method in our opinion results from the absence of studies on the impact of centrifugation itself on neutrophil functions.

The expression of CD11b, CD62L and CD66b on PMNs is typically regarded as an indicator of the cells’ activation status: While CD11b and CD66b are upregulated upon PMN-activation [18,28,29], CD62L levels decrease in response to neutrophil stimulation [30].

In our study, we first isolated neutrophil granulocytes from whole blood via centrifugation with different RCFs and then incubated the yielded cells with the stimulant fMLP for 24 h at 37 °C. Antigen expression was examined before (t = 0) and after (t = 24) incubation. CD11b expression levels at t = 24 were significantly lower, the higher the RCF applied during centrifugation. This was especially interesting regarding the fact that CD11b levels at t = 0—meaning immediately after centrifugation—did not significantly differ between the samples. In our opinion, this makes mechanical shredding, caused by centrifugation of surface antigens, rather implausible. While CD11b expression levels were still higher at t = 24 than at t = 0 for 9 k*g*s isolated cells, this was not the case for 18, 42 and 680 k*g*s cells, where expression levels were equal to (18 k*g*s) or even lower than before incubation (42, 680 k*g*s).

What we deduce from these findings is that the application of *g*-forces during centrifugation seems to inhibit the neutrophil granulocytes’ ability to upregulate CD11b expression in response to an activating stimulus. Since we only used fMLP as a stimulant in this setting, we cannot rule out the possibility of an affecting of the fMLP receptor or its downstream pathway being the reason for this inability. However, considering the already mentioned results of oxidative burst analysis, where PMA-stimulation led to ROS production equally as decreased as fMLP-stimulation, this explanation is not satisfying from our point of view. Furthermore, CD11b/CD18 is a key element of transepithelial [31] and transendothelial [32] migration of neutrophils. The inability of centrifuged neutrophils to upregulate CD11b expression after chemokine stimulation could be a possible explanation for why chemotactic migration is massively decreased in PMNs having experienced high *g*-times.

CD66b showed a similar—although not as distinct—picture as CD11b. CD62L was downregulated in all samples after 24 h of incubation with fMLP, regardless of the RCFs applied in the centrifugation process.

It seems that the effect of centrifugation has more influence on the expression of some surface antigens than on others. While significant differences in antigen expression after 24 h between different RCFs only occurred with CD11b and in parts with CD66b, there was no evidence of mechanical shredding of these surface proteins, since expression at t = 0 was relatively equal between the investigated RCFs for each of the three surface antigens analyzed. This alludes to the possibility of cell metabolic changes being induced by centrifugation. However, if and where there may be an impact of *g*-time on surface antigen expression on a protein translation level certainly needs further investigation.

In conclusion, the overall picture of reduced chemotactic migration, decreased oxidative burst and impaired upregulation, especially of CD11b expression, leads us to the assumption that the application of *g*-time may “paralyze” neutrophils. We deliberately describe our findings as a “paralytic” impact of centrifugation in this context, since—while the above-mentioned neutrophil functions are clearly impaired—we did not find any evidence of mechanical damage to the PMNs.

To our knowledge, spontaneous sedimentation (1*g*) has been compared to density gradient centrifugation only once by Mosca et al. [22]. However, this study—as described for the others before—featured centrifugation steps in both of the compared groups, which makes it inadmissible for actually assessing the influence of *g*-forces on neutrophils. We performed a literature analysis on different pre-existing comparisons of neutrophil isolation protocols and calculated a cumulative *g*-time load for each method used (see Table 3). The data from Sroka et al. [9] was particularly interesting. The “blood clot method” they used also avoids centrifugation and from this standpoint could be comparable to our 1*g* method. Furthermore, they described reduced chemotactic migration of neutrophils for every method they tested compared to the blood clot (or 1*g,* respectively) method. These findings would be comparable to the observations we made in our study.

However, there seems to be a general absence of research on the impact of centrifugation on any mammalian cells. In fact, most of the scant existing literature stems from the medical fields andrology and urology [33,34,35]. Interestingly, results of those studies on sperm cells are quite similar to our data on neutrophils: Marzano et al. were able to show, that with increasing *g*-force and duration of centrifugation, sperm cells suffered a major loss in motility compared to non-centrifuged samples [34]. Furthermore, Zini et al., as well as Marzano et al., observed a significant decrease in DNA integrity in centrifuged sperm cells in comparison to controls [34,35]. While we did not include examinations on a DNA level in our study yet, this may be a starting point for future investigations on the effects we discovered in neutrophils.

Our results implicate that previous studies, which have had centrifugation steps in their sample preparation procedure, might have been performed with already drastically impaired neutrophils. While we do not doubt the validity of said studies and results, we assume that even greater and more distinct effects could have been found if centrifugation had been avoided beforehand. For future investigations, we strongly advise researchers to prepare their neutrophils with as minimal *g*-time stress as possible.

Limitations: Since the examination of surface antigen expression via flow cytometry required a high cell purity, two centrifugation steps immediately before measurement were necessary during sample preparation. The combined *g*-time load of those steps was 209 k*g*s. Those were performed at a temperature of 4 °C in order to temporarily down-regulate cell metabolism. While we believe to have taken enough measures against impairment of our neutrophils (equal treatment of control and non-control cells; centrifugation at 4°C only immediately before measurement), we cannot entirely rule out an interference with our results in this specific case.

Both of our burst (ROS production) results with the highest *g*-time stress for PMNs seem to be contradictory to the results from the less stressed cells. As described in the methods section, the 756*g* centrifugation was conducted in density gradient medium. In an earlier publication of our department, we were able to show that the PMN-carrying matrix itself has an influence on the cells’ functions [36]. This might be the reason for the contradictory results.

The number of live cell imaging tests (n = 7) is limited. However, since this method delivered particularly clear results, we regard the number of tests as sufficient. The number and kind of surface proteins we investigated are also limited. More holistic approaches (e.g., metabolomic and proteomic analyses or investigations on changes of the cytoskeletal architecture) should follow this initial project.

## 5. Conclusions

Centrifugation significantly impairs major neutrophil functions. Chemotactic migration and oxidative burst are significantly decreased in centrifuged neutrophils. Upregulation of CD11b and, in part, CD66b expression is partly inhibited in PMNs having experienced centrifugation. Those effects are dependent of the *g*-time applied, a parameter combining the two variables *g*-force and centrifugation duration. Correlation between *g*-time and neutrophil impairment was sigmoidal in the *g*-time ranges we examined. Further investigation on how exactly *g*-time impairs those neutrophil functions on a protein or molecular level is necessary. With centrifugation being a part of nearly every neutrophil isolation protocol, previous studies may not have produced the desired effects as they may have worked with drastically impaired PMNs. Future studies should avoid centrifugation entirely if possible or at least reduce the *g*-time load which neutrophils have to endure during the isolation process.

The overall picture of reduced chemotactic migration, decreased oxidative burst and impaired upregulation of surface antigen expression leads to the assumption that centrifugation induces a *g*-time-dependent “paralysis” of neutrophil granulocytes.

## Figures and Tables

**Figure 1 biomedicines-10-02896-f001:**
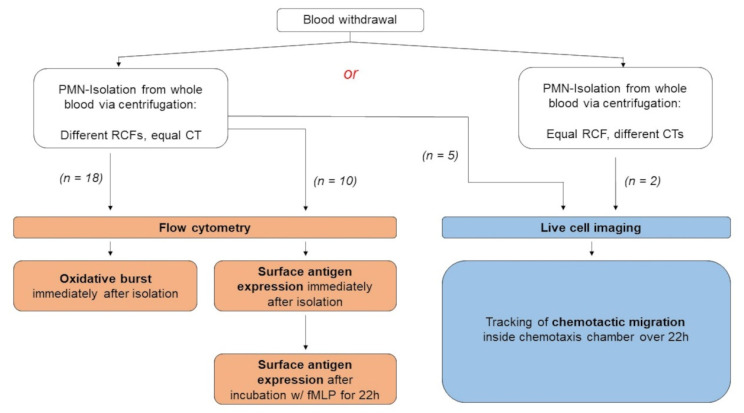
Schematic presentation of the methods used in this study. In one setup, centrifugation is performed with different relative centrifugal forces (RCFs) but one centrifugation duration (CT). In the other setup, centrifugation features equal RCF but different CTs. For examination of surface antigen expression, isolated cells are incubated with chemoattractant n-formylmethionyl-leucyl-phenylalanine (fMLP).

**Figure 2 biomedicines-10-02896-f002:**
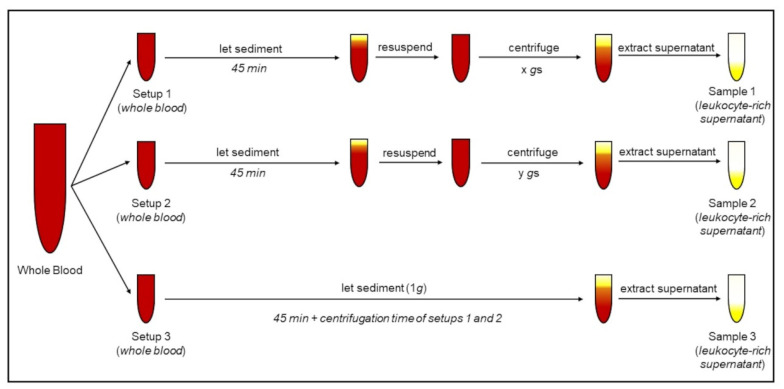
Protocol for isolation of polymorphonuclear leukocytes (PMNs) from whole blood. “Setup 3” represents the control setup, where isolation is performed without centrifugation (1*g*). “Setup 1” and “Setup 2” represent samples centrifuged with different *g*-times.

**Figure 3 biomedicines-10-02896-f003:**
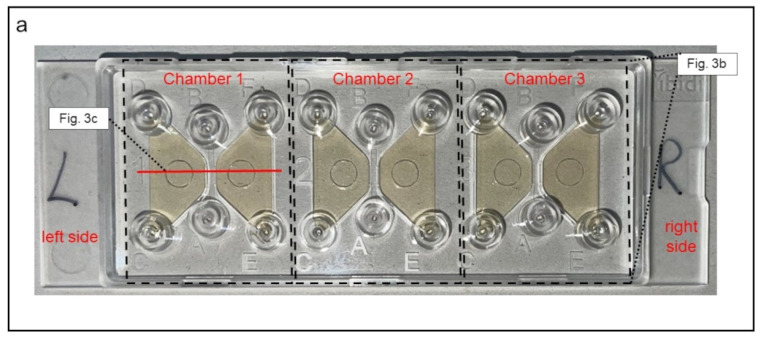
Demonstration of the 3D-µ-Slide migration-chambers used for analysis of neutrophil migration. (**a**) Picture of the slide. The three different chambers are each framed (dashed lines). In each chamber, both reservoirs are colored yellowish for clarification. (**b**) Schematic display of one chamber from top view. The central channel is positioned between both reservoirs in each chamber. (**c**) Cross-sectional view of one chamber. The right reservoir contains the leukocyte-suspension, whereas the left reservoir serves as source for the chemoattractant fMLP. The central channel is filled with a collagen matrix that migrating neutrophils must pass. C_100_(fMLP) indicates that 100% of the chemoattractant is located in the left reservoir immediately after the filling. C_0_(fMLP) indicates that no chemoattractant is located in the right reservoir immediately after the filling.

**Figure 4 biomedicines-10-02896-f004:**
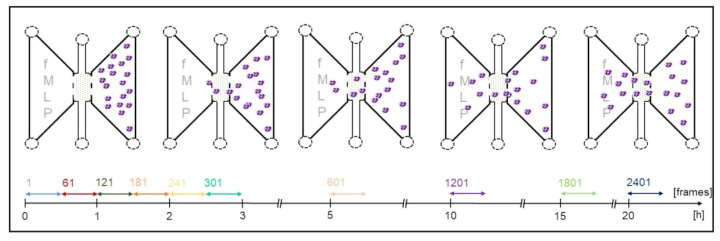
Correlation between analysis interval, start frame and time of observation. Each double arrow represents one analysis interval (=60 frames) and is labeled after its first frame (=start frame).

**Figure 5 biomedicines-10-02896-f005:**
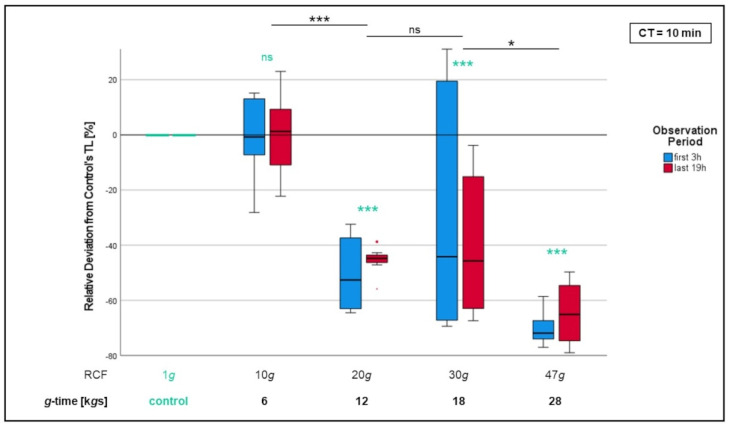
Comparison of chemotactic migration of PMNs after isolation with different RCFs but an equal CT of 10 min. Mean track length of 1*g* control cells was set as default value (“0”; turquoise colored). Mean track lengths of non-control cells are expressed as “relative deviation from control’s track length (TL)”. Significant differences compared to the 1*g* control are turquoise colored. Significant differences between non-control samples are indicated with bars. Differences between analyzed groups in graphics are marked as: ns = not significant; * *p* < 0.05; *** *p* < 0.001.

**Figure 6 biomedicines-10-02896-f006:**
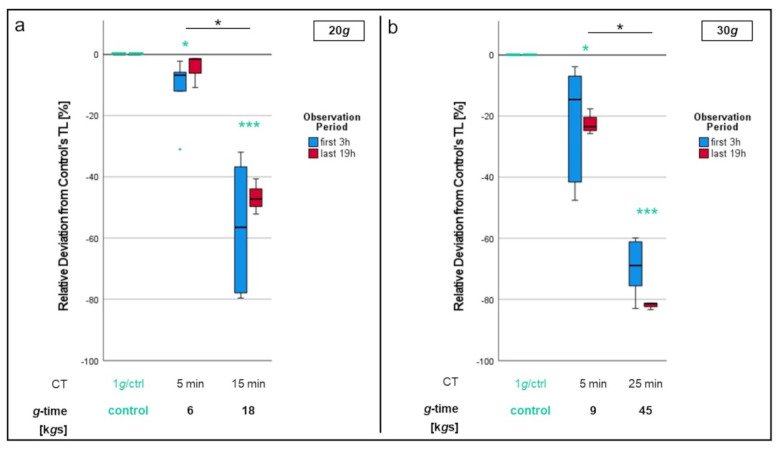
Comparison of different centrifugation durations for 20*g* (**a**) and 30*g* (**b**) regarding neutrophil migration. Significant differences compared to the 1*g* control are turquoise colored. Significant differences between non-control samples are indicated with bars. Differences between analyzed groups in graphics are marked as: * *p* < 0.05; *** *p* < 0.001.

**Figure 7 biomedicines-10-02896-f007:**
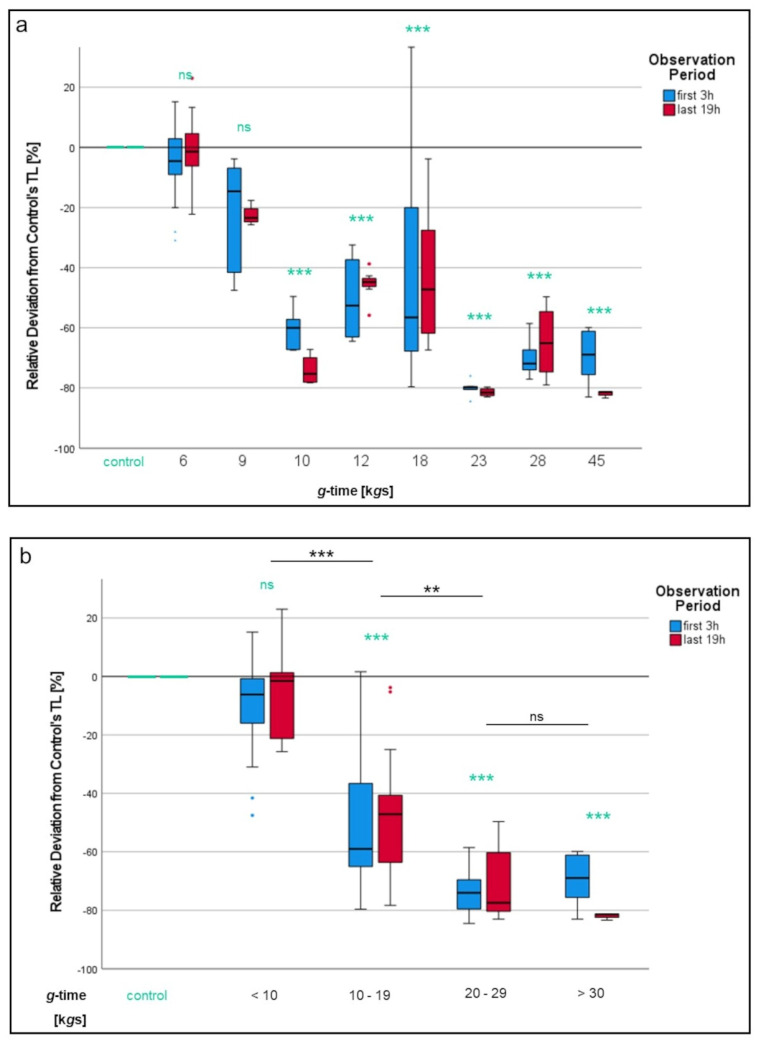
Impact of *g*-time on neutrophil migration. (**a**) Each *g*-time group in detail. In (**b**), groups are summarized into different *g*-time intervals. Significant differences compared to the 1*g* control are turquoise colored. Significant differences between non-control samples are indicated with bars. Differences between analyzed groups in graphics are marked as: ns = not significant; ** *p* < 0.01; *** *p* < 0.001.

**Figure 8 biomedicines-10-02896-f008:**
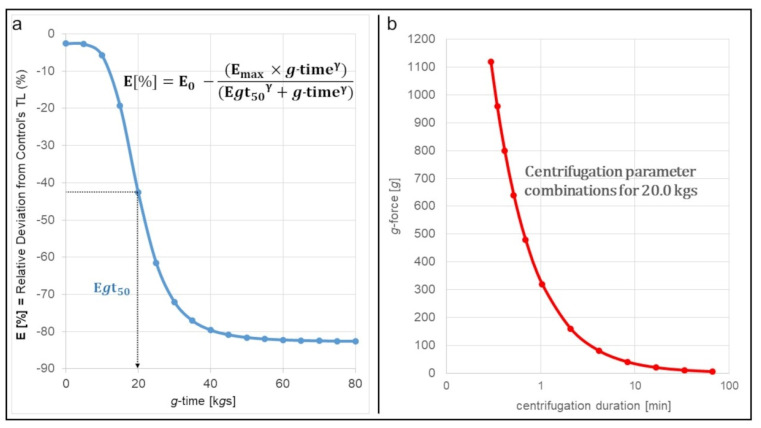
Graphical presentation of the correlation between *g*-time and relative deviation from control’s track length. (**a**) Inhibitory dose–response curve for the effect of *g*-time on neutrophil migration. “E” represents the effect size, in this case the rTL-value. “E_0_” represents the initial level. “E_max_” represents the maximum effect size. “E*g*t_50_” indicates the *g*-time value, at which half of the maximum effect has been reached. (**b**) Possible combinations of centrifugation parameters to reach the E*g*t_50_ value of 20.0 k*g*s.

**Figure 9 biomedicines-10-02896-f009:**
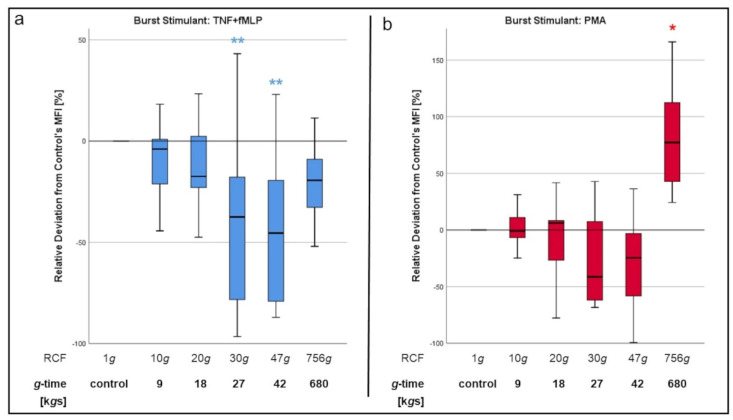
Oxidative Burst of PMNs after stimulation with tumor necrosis factor alpha (TNF-α) + fMLP (**a**) or phorbol-12-myristate-13-acetate (PMA) (**b**). Significant differences compared to the 1*g* control are turquoise colored. Significant differences between non-control samples are indicated with bars. Differences between analyzed groups in graphics are marked as: * *p* < 0.05; ** *p* < 0.01.

**Figure 10 biomedicines-10-02896-f010:**
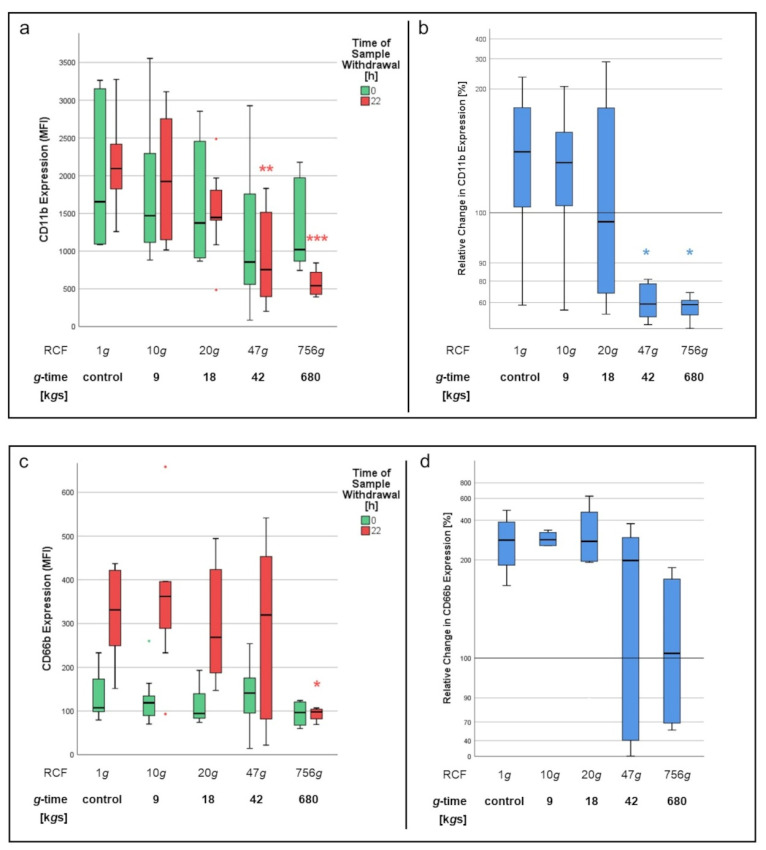
Expression of surface antigens CD11b (**a**,**b**) and CD66b (**c**,**d**) before (t = 0) and after (t = 22) incubation with fMLP. Significant differences (* *p* < 0.05; ** *p* < 0.01; *** *p* < 0.001) are determined as compared to the 1*g* control at the same time of measurement. Colorings indicate the time of sample withdrawal. In (**b**,**d**), expression levels at t = 0 were determined as a relative change value = 100. Therefore, relative change values of >100 indicate increased expression at t = 22, whereas values of <100 indicate decreased expression at t = 22.

**Table 1 biomedicines-10-02896-t001:** Composition of the 3D-µ-Slide’s compartments. More detailed information about the components can be found in Table A1 in the Appendix A.

Left Reservoir (65 µL)	Central Channel (6.5 µL)	Right Reservoir (65 µL)
fMLP [10 nM] in autologous serum	Medium (33%)	Leukocyte-rich supernatant (50%) Autologous serum (50%)
	RPMI 1640 (15%)	
	Autologous Serum (1.7%)	
	Collagen Type I (50%)	

**Table 2 biomedicines-10-02896-t002:** Parameters used for examination of neutrophil migration. Parameters are defined according to [5].

Parameter	Unit	Description
Track displacement (X, Y)	[µm]	Distance, which tracked cell has traveled in (X) and vertically to (Y) direction of chemoattractant
Track displacement length	[µm]	Euclidean length of the straight line between starting point and end of the track
Track duration	[s]	Timespan, in which single cell was tracked
Track length	[µm]	Distance, which tracked cell has covered
Track speed	[µm/s]	Speed, with which tracked cell has traveled
Track straightness	range between 0–1	Degree, to which a tracked cell travels on a straight line or deviates from it (1 = maximal straightness)

**Table 3 biomedicines-10-02896-t003:** Analysis of literature regarding the *g*-time load of different isolation protocols. The method with less *g*-time load was set as the reference method in each case. Abbreviations: DG = density gradient separation; IM = immunomagnetic separation; SPO = spontaneous sedimentation; CL = blood clot method; AC = ammonium chloride method; CM = chemotactic migration; ROS = s oxidative burst. “↑” indicates higher, “↓” indicates lower, “↔” indicates no change in functions compared to the reference method. Green shaded table elements indicate accordance to our results (“higher *g*-time impairs neutrophil functions”). Since the reference method (blood clot method) in [9] did not include a centrifugation step, we denominated it as “1*g* control”.

Literature	Reference Method	*g*-Time Load Reference Method [k*g*s]	Test Method	*g*-Time Load Test Method [k*g*s]	Neutrophil Functions Compared to Reference Method	Comments
[13]	IM	84	DG	741	CM ↔; ROS ↓	
[22]	SPO	126	DG	621	CM ↓	

[22]	DG	621	DG	954	CM ↔	Comparison of very high *g*-times
[9]	CL	1*g control*	AC	97	CM ↓	
[9]	CL	1*g control*	DG	882	CM ↓	
[9]	CL	1*g control*	DG	1266	CM ↓	
[8]	IM	570	DG	1600	ROS ↔	
[23]	IM	840	DG	>1000	ROS ↔ -↓	Only eosinophils were examined
[24]	DG	624	DG	1580	CM ↓	
[25]	AC	240	DG	1500	CM ↓; ROS ↑	Only canine neutrophils were examined

## Data Availability

The data presented in this work are available on request from the corresponding author.

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
