# Peer review of "Paralytic Impact of Centrifugation on Human Neutrophils"

_biomedicines, 2022, doi:10.3390/biomedicines10112896_

Round 1

Reviewer 1 Report

Purification of neutrophils with different known methods is known to modify its functional characteristics. This work was performed to explore the effect of centrifugation force and time on the function of neutrophils. The study is well designed, and the different methods are properly explained in text and with the help of the figures.

Seeing that the neutrophils function better when harvested using smaller centrifugal forces and centrifugation times, it seems like a small sacrifice to make this a regular practice in studies using neutrophils and the overall results make this study useful.

Reviewer 2 Report

The authors investigated the effect of centrifugation on human neutrophils.

Major:

The experiment design, in my opinion, is flawed. The authors compare Human Neutrophil isolated under various conditions. The population of isolated cells may differ. Figure 10a shows that separated neutrophils had distinct CD11b before the experiment. The authors should first separate neutrophils, then centrifuge the same batch of cells under varied conditions to evaluate neutrophil function.

Reviewer 3 Report

The manuscript submitted by Authors deals with the paralytic impact of centrifugation on human neutrophils. Overall, the manuscript is really interesting. Congratulations to the Authors of the topic and interesting results.

I just have a few questions that I would like answers to:

1. There are incorrect captions in Figure 3a. Instead of Figure 5b and 5c, it should be 3b and 3c. Please correct this.

2. Why didn't the Authors consider standard cell staining (e.g. using trypan blue which is used to document cell death) in the paper, as is done in in vitro laboratories ?

3. In general, neutrophil isolation is usually used in scientific research. Nevertheless, lymphocyte isolation is most often used in laboratory diagnostics. In our laboratory, we perform lymphocyte isolation, which is then used for serological typing in donor and recipient selection. Can the Authors address this issue in their future research? Often when performing a lymphocytotoxic test on Terasaki plates, it is apparent that lymphocytes are recognized by antibodies and then destroyed by complement. This may also be related to the isolation methodology using a density gradient and centrifugation. I encourage the Authors to perform such studies in the future.

Round 2

Reviewer 2 Report

I'd like to accept the rebuttal and congratulations on your excellent work.